# Person-to-person opinion dynamics: An empirical study using an online game

**Johnathan A. Adams** [1], **Gentry White**[1,2], **Robyn P. Araujo**[1,3]*

**1** School of Mathematical Sciences, Queensland University of Technology, Brisbane, Queensland, Australia,
**2** QUT Centre for Data Science, Queensland University of Technology, Brisbane, Queensland, Australia,
**3** Institute of Health and Biomedical Innovation, Kelvin Grove, Queensland, Australia

* r.araujo@qut.edu.au

## Abstract

A model needs to make verifiable predictions to have any scientific value. In opinion dynamics, the study of how individuals exchange opinions with one another, there are many theoretical models which attempt to model opinion exchange, one of which is the Martins model, which differs from other models by using a parameter that is easier to control for in an experiment. In this paper, we have designed an experiment to verify the Martins model and contribute to the experimental design in opinion dynamic with our novel method.

## Introduction

The field of opinion dynamics has a wide variety of theoretically derived models that potentially describe human interactions and the resulting change in opinions. Despite the appeal of these models, there is a dearth of empirical evidence to support their utility [1]. For a model to be scientifically verifiable, the model needs to make testable predictions about the outcome of an experiment. While modern examples of research [2, 3] demonstrate an effective method to investigate opinion dynamics models, most theoretical opinion dynamics models don't offer predictions on behaviours, which makes it challenging if not impossible to create controlled experiments which can verify these models [4]. Consider the bounded confidence model [5, 6], which includes the parameter $\epsilon$ limiting agent interactions. Certain values of $\epsilon$ can create polarisation. But because $\epsilon$ is an abstract (and highly subjective) measure in opinion space, it is difficult to create an experimental condition to control $\epsilon$. In general, the level of abstraction in the opinion dynamics models' parameterisations limits the design and implementation of experiments for testing model validity. Further, opinion dynamics models created from data are also difficult to verify because, as stated in [4], models fitted to the data of experiments rarely make testable predictions about future data. We break this trend by designing and executing an experiment testing the claims made by the Martins model [7].

The Martins model [7] represents opinions as probability density functions such that a person has an opinion $x \in \mathbb{R}$ and an associated uncertainty $\sigma \in \mathbb{R}^+$. Their opinion and uncertainty represent a Gaussian density function with mean $x$ and standard deviation $\sigma$. When two agents interact in the Martins model, they share their opinions (and in the extended model [8] their uncertainties), and the two agents then update their opinion and uncertainty via Bayesian

**Data Availability Statement:** All relevant data are within the paper and its Supporting information files.

**Funding:** Robyn P. Araujo is the recipient of an Australian Research Council (ARC) (https://www. 

arc.gov.au/) Future Fellowship (project number FT190100645) funded by the Australian Government. The funders had no role in study design, data collection and analysis, decision to publish, or preparation of the manuscript.

**Competing interests:** The authors have declared that no competing interests exist.

updating. A key parameter in the model is $p \in [0, 1]$, which is the propensity for agents to believe that other agents have useful information and use that information to update their opinions. When $p = 1$ consensus is always reached, whereas values of $p < 1$ polarisation emerges. Compared to the bounded confidence model's $\epsilon$, the parameter $p$ is much more interpretable and controllable (in an experimental setting) than the parameter $\epsilon$.

We present, in this article, a comprehensive literature review of previous empirical studies in opinion dynamics. Then we outline a design for an experiment which can test whether the Martins model can predict opinion shifts of individuals. We executed such an experiment and, in this article, present the results of the experiment. In the results, we found two distinct phenomena occurring in the experiment: when two individuals are close in opinion, the Martins model made a reasonable prediction of the opinion shift; when two individuals are far in opinion, the observed opinion shift followed a what would be expected from discrete opinion choice model. We concluded by discussing our novel results and identifying the limitations of our experiment.

## Previous experiments

There is limited evidence of direct use in opinion dynamics of experimental data to either verify hypotheses based on model predictions or construct empirical models. This scarcity of evidence is partly due to the difficulty of designing an experiment that accurately replicates real-world interactions while controlling the experimental conditions and has resulted in empirical data collection in opinion dynamics evolving independently from the theory.

## Experimental data collection

Many empirical investigations into opinion dynamics draw inspiration from psychology studies that investigated opinion change [9–12]. All of these studies served as guidelines for the experimental design of the later opinion dynamics studies. For example, the study [11] aimed to test two hypotheses: "Extreme members will contribute more to the group discussion than less extreme members. (1a) They will use more words than less extreme members, and (1b) they will take more turns than the less extreme member," and (2) "There should be greater group polarisation in a group containing an extreme member than in groups not containing an extreme member." The authors of [11] tested these by dividing 129 participants into 43 groups of three. Participants in each group were asked their opinion and knowledge on the legalisation of marijuana before the experiment. The participants read material related to the legalisation of marijuana and then discussed the issue within their group until the group reached a compromise. After the discussion, participants reevaluated their opinion. This experimental design formed the bases for the approach to collecting data in the opinion dynamics literature.

## Building empirical models

While Opinion Dynamic's inception began in the 1950's [13], one of the first significant studies focused on developing a model of opinion change using experimental data was published in 2013 [14]. The experimental design in [14] draws direct inspiration from the previous psychology literature, but the study generated a model of human behaviour from the collected data rather than prove any specific hypothesis. Participants were asked general knowledge questions with a real number answer, e.g. "How long is the Mississippi river?" and rated their confidence in their answer on a 1 to 6 scale, with lower meaning less confident. Participants only saw one other participant's answer and confidence at a time.

The authors of [14] used the experimental data to create an influence map of the experimental subjects' behaviours. An influence map is a surface in relative opinion and relative uncertainty space, which describes the opinion change of an individual according to their relative opinion and relative uncertainty with a hypothetical interaction partner. The authors used the influence map to create a decision tree model. Depending on where an interaction fell on the influence map, the model specifies three ways an agent could update their opinion after an interaction with another agent: *rejecting* where there is no change in opinion; *compromising* where the opinion shifted 'halfway' towards the other opinion; and *adopting* where the opinion changed to be the other opinion. The resulting model is related to the bounded confidence model [5, 6] such that the regions of rejecting, compromising and adopting could be used to determine an $\epsilon$, but the influence map of [14] implies a more nuanced picture which the bounded confidence model cannot address.

More modern models like the Martins [7] and relative agreement models [15] produce similar behaviour seen in the influence map generated by [14]. But it is difficult to precisely confirm whether the models like Martins or relative agreement can accurately predict the behaviour observed in [14]. Specifically, the Martins and relative agreement models rate confidence/uncertainty as continuous values, which conflicts with the discrete 1–6 scale [14] used to measure confidence, therefore making the empirical data incompatible with the theoretical models. The goal of [14] was to use the data to generate a model, not to verify an existing model.

## Other experimental designs

Modern studies have improved the experimental design of [14]. For example, the work [3] provided a novel contribution where participants interacted in pairs through digital displays and exchanged their opinions, but participants could see their interaction partner update their opinion in real-time. The new method proved a controlled, yet realistic, environment to test ideas about opinion exchange and revealed new behaviour in which participants became more confident upon observing that their interaction partner changed their opinion. This empirical evidence provides clues to produce theoretical models which can predict opinion exchange more effectively.

More significant is the work of [2]. Specifically, the study [2] investigates how groups assessed threatening objects and developed a model similar to the French model [13] which establishes seven testable predictions ranging from conditions on how individuals modify their threat assessments to predicting the process in which society might reach consensus. Agents in the model of the study, like in the original French model [13], weighted their neighbours such that when the model progressed, the agents would adopt the weighted average opinion of their neighbours according to the weighting the agent assigned each neighbour. For example, consider a three agent simulation with Agents 1, 2 and 3 holding opinions $x_1$, $x_2$, $x_3$ respectively, and Agent 1 weighting every agent in the simulation according to this vector [0.2, 0.3, 0.5]. When the model updates Agent 1's opinion will be $0.2x_1 + 0.3x_2 + 0.5x_3$. The study measured these weightings by first giving 100 chips to each participant after the group discussion. Next, participants distributed their chips according to how much they were convinced by other group members that an object was "threatening". Participants were allowed to keep chips if they were not convinced by the group. The chip distribution provided by each participant directly measured the weightings necessary for the modified French model. The study concludes by evaluating the model's ability to predict an individual's threat assessments. The work of [2] demonstrates an effective method to evaluate the opinion dynamics model, which this paper hopes to emulate.

The side of opinion dynamics concerned with measuring the most influential individual has already produced empirical studies that seek to verify theoretical models' predictions. The work of [16] offshoots from the French model [13] by imposing that the weights of the French be related to the in-degree agents, i.e. how well listened an agent is. This weighting scheme relies on a parameter $\rho$ such that when $\rho = 0$ in-degree does not affect opinion dissemination when $\rho = 1$ neighbours are weighted proportionally to an agent's personal in degree and when $\rho \rightarrow \infty$ agents only listen to the neighbour with the most in-degree. In the study, [17] the authors developed an experiment that isolated the effect of in-degree. Specifically, the authors developed a social network for participants that controlled for in-degree. The result of the experiment was the rejection of the null hypotheses, i.e. $\rho = 0$, suggesting that in-degree has a role in opinion dissemination. With the experiment of this study, we seek to accomplish a similar goal on the inter-personal level and ascertain whether mistrust influences interpersonal communication as the Martins model describes.

## Materials and methods

In comparison with previous work, we designed this study's experiment to be more abstract. This abstraction allows for a more direct comparison between the model variables, i.e. agent uncertainty and opinion, and the data collected from the experiment. In addition, the abstraction minimises the impact of cultural bias. Consider the general knowledge question used in previous experiments. The questions limited the pool of participants to those somewhat knowledgeable of the topic, e.g. a question like "How long is the Mississippi river?" limits participants to those from the US. We naturally avoided this problem with our experiment. We took advantage of this flexibility to make the study a snowball sample study; participants are encouraged to invite others to participate, to increase the sample size for the study.

### Ethics statement

The following experimental design and experiment was approved by the Queensland University of Technology (QUT) Human Research Ethics Committee (UHREC) as Negligible-Low Risk. Reference number: 2000000739.

### Recruitment of participants

As stated previously, our recruitment strategy for this experiment was a snowball sampling strategy. We advertised the experiment on the social media websites Facebook and YouTube and through the mailing list and Slack workgroups of the QUT mathematics school. When a participant finished the experiment, we recorded the IP address associated with the device they used for the experiment as part of the data. We recorded IP addresses to determine the number of unique participants in the experiment. We recorded no other personal data on the participant. A total of 257 unique participants participated in the experiment, assuming that participants are unique to each IP address.

### The experiment

The experiment entailed playing a game on the internet hosted on QUT servers. The goal of the game was to find a hidden dot inside a black box on screen. Participants were given information to find the dot in the context of a social interaction. Participants would first see a blue circle which was explained as information that was always reliable. The hope being that a participant would internalise the blue circle as their opinion. Next a participant would see a red circle which was explained as not being reliable all the time. The idea being that participants

would interpret the red circle as rumours which are subject to being false. Lastly a participant would draw a new circle in response to the red and blue circles. We directly controlled the reliable of the red circle while informing the participants which allowed us to control for $p$ in the Martins model. See S1 File for the source code of the website and in total we had 3760 games played.

**Participant instructions.**   Before any participant played a game, they first saw instructions for the game. The instructions described the game similarly to how it is described in this paper except without the probability terminology, e.g. confidence intervals, to avoid confusion for the participants. The instructions developed a backstory to the game to encourage participants to role-play so that the participants responded realistically. The instructions described an eccentric Flemish trillionaire, Monsieur Dotte, as hosting the game, and they wanted the world to indulge in his passion for puzzles and social deduction. So, M. Dotte offered a 'cash prize' for those who do well at his game of finding the dot. This framing allowed us to communicate specific information to participants, e.g. the reliability of the red circle at different traffic light signals and the 80% chance a reliable circle had in containing the dot, while keeping the scenario plausible in the minds of the participants. It was made clear that there was no monetary reward for playing the game and the 'cash' was just their score after finishing the game and held no fiscal value. See S1 Fig for the instructions we gave to the participants on the game website.

**The game.**   Initially, a participant would see an empty black box. The participant would then click on the box resulting in the blue circle appearing, e.g. they could see S2 Fig. The blue circle represented an 80% confidence interval which was explained to the participants as an 80% to contain the dot. The blue circle was 100% reliable and always gave information on the dot's location. The dot could be outside the blue circle, but because the blue circle was 80% confidence interval, the dot would appear close to the circle.

When participants clicked again, the red circle would appear, e.g. they could see S3 Fig. Like the blue circle, the red circle purports to be an 80% confidence interval of the dots' position, but the red circle has a probability of being unreliable, i.e. drawn at random and independent of the dot's actual location. To communicate the unreliability of the red circle a traffic light above the box would light up such that: when the traffic light was red, the red circle had a reliability of 20%; when the light was yellow, the red circle had a reliability of 50%; when the light was green, the red circle had a reliability of 80%. The confidence interval of the red circle remained constant when the red circle was randomly determined to be reliable. Otherwise, the red circle would be drawn randomly inside the box. These reliability probabilities were communicated to the participants in the instruction and were our attempt to exactly quantify $p$ to allow for more definitive model predictions.

Finally, the game directed the participant to consider the position and reliability of the circles and draw a new circle (hereafter, the 'user circle') that they believed contained the dot. After the participant finished drawing the user circle they were scored based on their accuracy (whether the dot was in their circle) and their precision (how small their circle was) relative to the blue circle and were encouraged to play again. Then we recorded the final game states, including the size and position of all three circles, the reliability of the red circle, a unique session id and the participant's IP address. S4 Fig shows an example of a finished game.

**Collected data.**   The Martins model [7] and its extension [8] predicts an individual's shift in opinion based on the parameter $p$, the opinions $x$ and the uncertainties $\sigma$ of both individuals involved in an interaction. As part of the experiment we recorded the specific values for $p$, $x_i$, $x_j$, $\sigma_i$ and $\sigma_j$ of every simulated interaction. i.e. every game that a participant played and Table 1 describes how we organised that data.

**Table 1. Data collected from the experiment.**

| Variable name | Description |
|---|---|
| $p$ | The probability of the red circle giving useful information |
| $x_{blue}$ | The x-coordinate of center of the blue circle |
| $y_{blue}$ | The y-coordinate of center of the blue circle |
| $r_{blue}$ | The radius of the blue circle |
| $x_{red}$ | The x-coordinate of center of the red circle |
| $y_{red}$ | The y-coordinate of center of the red circle |
| $r_{red}$ | The radius of the red circle |
| $x_{user}$ | The x-coordinate of center of the user circle |
| $y_{user}$ | The y-coordinate of center of the user circle |
| $r_{user}$ | The radius of the user circle |

The blue circle is the first circle seen by the participant and represents the knowledge already acquired. The red circle is the second circle seen by the participant and represents the opinion and conjecture of another actor. The user circle is the circle drawn by the participant.

Except for $p$, the data in Table 1 are in units of pixels, whereas the Martins model deals in the abstract (unitless) opinion space. For clarity and similitude, we scaled all the relevant data removing the unit of pixels. We calculated the scaling factor by finding the radius of the circle of area equivalent to an HD monitor display (1920 by 1080). We then divided the radius by the number of standard deviations to produce an 80% confidence interval, resulting in 634 pixels per standard deviation. We used this factor to scale the data and remove the unit pixels.

**Scoring.** We encouraged participants to play multiple games by giving the participant a score per attempt. Scoring a participant's game follows these steps:

1. Calculate an accuracy $A_{user} \in \mathbb{R}^+$ and precision $P_{user} \in \mathbb{R}$ rating for the participant based on the circle they drew. Note that a negative value for precision results when the area of the player's circle approaches the area of the box.

2. Produce an overall rating for the participant, $R_{user}$, as a weighted sum of $A_{user}$ and $P_{user}$ with weights $w_A$ and $w_P$, respectively, e.g. in the experiment $w_A = 0.1$ and $w_P = 70$ are chosen based on preliminary experimentation to determine intuitive scoring results.

3. Repeat steps 1 and 2 for the blue circle, producing a rating for the blue circle $R_{blue}$.

4. Find the relative rating $R$ for the participant's circle

$$R = R_{user} - R_{blue} + R_0,$$

where $R_0$ is the rating given for guessing equally well as the blue circle, e.g. for the experiment $R_0 = 2 \times 10^{-3}$. Giving a participant a rating relative to the blue circle encourages participants to guess better than the guaranteed information they start with.

5. Calculate a score $S \in [0, S_{max}]$ as a sigmoid function of $R$, e.g. the experiment used

$$\frac{S_{max}}{1 + e^{-R/2}}.$$

The constant $S_{max}$ is the maximum score achievable when playing the game, e.g. in the experiment $S_{max}$ is one hundred thousand.

*Remark.* To calculate accuracy and precision we used the following

$$A(e, r) = \frac{1}{1/L_{\max} + e} + U_{\text{bonus}} \frac{r - r_{\min}}{r_{\min}}, \tag{1}$$

$$P(r) = P_{\text{factor}} \frac{1}{\pi(r - r_{\min})^2} - \frac{\pi(r - r_{\min})^2}{B_{\text{area}} - \min[B_{\text{area}}, \pi(r - r_{\min})^2]}, \tag{2}$$

where $e$ is the error of the circle which is the distance between the centre of the circle and the dot, $r$ is the radius of the circle, $r_{\min}$ is the radius of the smallest possible circle that can be drawn, $L_{\max}$ is the maximum accuracy score achievable when getting $e = 0$, $U_{\text{bonus}}$ controls how much the circle radius factors into accuracy, $B_{\text{area}}$ is the area of the box containing the dot and $P_{\text{factor}}$ controls for how precise a circle of a given area is. For the experiment $L_{\max} = 100$, $U_{\text{bonus}} = 0.01$ and $P_{\text{factor}} = 1$.

We rated accuracy and precision this way for two main reasons. First was so that accuracy and precision would be completely unrelated to the Martins model because if it were, participants would be encouraged to guess more like the Martins model, thus biasing the data. The second was for the ratings to produce a "fair" score by relating it to tangible concepts, e.g. a circle the size of the box would be considered very imprecise and thus would give no score, i.e.

$$P_{\text{user}} \to -\infty \Rightarrow R_{\text{user}} \to -\infty \Rightarrow S = 0.$$

A score that a participant considers fair would encourage them to continue playing, at the very least not dissuade them.

## Model and data predictions

The Martins model predicts that a user's opinion will fall on the line segment connecting the centres of the blue and red circles. The predicted distance from the centre of the blue circle to the user's opinion is

$$h_{\text{expected}} = p^* \frac{d}{1 + R_\sigma^2}, \tag{3}$$

where $d$ is the distance between the centres of both the blue and red circle, $R_\sigma$ is the ratio of both the red and blue circles' confidences which in this case means the ratio of both circles' radiuses and $p^*$ is a variable in the model dependent on $d$ relative to $R_\sigma$ (see S1 Appendix for more details). Similarly, the difference between the variances of the user circle and the blue circle is predicted to be

$$k_{\text{expected}} = p^* \left( \frac{1}{1 + R_\sigma^2} \right) \left( (1 - p^*) \frac{(d)^2}{1 + R_\sigma^2} - \sigma_i^2 \right). \tag{4}$$

We can multiply this quantity by $\pi$ and $1.29^2$ to get the expected change in circle area, where 1.29 is the number of standard deviations away from the mean required to construct an 80% confidence interval. These theoretical values can be directly compared to the observed data and assessed for the goodness of fit.

We calculate the observed shift towards the red circle $h$ and change in circle area $k$ as

$$h_{\text{observed}} = \frac{(x_{\text{blue}} - x_{\text{user}})(x_{\text{blue}} - x_{\text{red}}) + (y_{\text{blue}} - y_{\text{user}})(y_{\text{blue}} - y_{\text{red}})}{d}, \tag{5}$$

$$k_{\text{observed}} = (r_{\text{blue}}^2 - r_{\text{user}}^2)\pi. \tag{6}$$

To compare the Martins model with the data, we calculate the following: $d$, the distance between $x_i$ and $x_j$; $\sigma_i$, the uncertainty of initial belief; $R_\sigma$, the ratio of $i$ and $j$'s confidences; and the Martins model quantity $p^*$. The value $d$ is the distance between the centers of both the blue and the red circles and is thus

$$d = \sqrt{(x_{\text{blue}} - x_{\text{red}})^2 + (y_{\text{blue}} - y_{\text{red}})^2}.$$

The circles are presented to the participants as 80% confidence intervals, therefore

$$\sigma_i = r_{\text{blue}}/1.29$$

where 1.29 is the number of standard deviations from the mean required to get 80% confidence. The ratio of $i$ and $j$'s confidences is

$$R_\sigma = \frac{r_{\text{blue}}}{r_{\text{red}}}.$$

The quantity $p^*$ is a function of $d$, $\sigma_i$ and $R_\sigma$ and is

$$p^* = \frac{p\phi(d, \sigma_i\sqrt{1 + R_\sigma^2})}{p\phi(d, \sigma_i\sqrt{1 + R_\sigma^2}) + (1 - p)} \tag{7}$$

where

$$\phi(d, \sigma_i\sqrt{1 + R_\sigma^2}) = (1/(\sigma_i\sqrt{2\pi(1 + R_\sigma^2)}))e^{-(d)^2/2\sigma_i^2(1+R_\sigma^2)}. \tag{8}$$

Intuitively these predictions mean, even when $p$ is low, we expect to see participants shift towards the red circle more and reduce the size of their circle (relative to the blue) more when the red and blue circle are close to each other. Likewise, even when $p$ is high, we expect to see participants remain close to the blue while keeping the same radius as the blue circle when the red and blue circles are very far apart. This is due to the influence of $p^*$, because $p^*$ controls the degree an agent incorporates a new opinion and $p^*$ depends on $d$, i.e. the difference in opinion, and $\sigma_i^2 + \sigma_j^2$, i.e. the total variance of both agents' opinion. The reliability $p$ only effects the speed at which agents in the model effectively trust other agents and hence doesn't produce significantly different behaviour for values of $p$ between 0.2 and 0.8 (except for extreme case like when $p \to 1$ [8]). Given the Martins model prediction on the circle participants draw in the game we supply the hypotheses

1. There is no correlation between the observed and expected shifts away from the blue circle.

2. There is no correlation between the observed and expected change in circle area from the blue circle.

## Results

For each game, we calculated the user's predicted shift from the blue to the red circle and the change in their circle area compared to that of the blue circle using the Martins model. We compared the predicted results with the observations in Fig 1 (for the raw dataset see S2 File). Upon initial observation, we see in the data a few outliers where participants drew large circles in random places relative to the blue circle. We surmise that participants were likely trying to 'break' the game in the experiment by drawing the largest circle possible. More interestingly, we can see two patterns emerge from the data. First is the linear relationship we expect to see between the observation and what the Martins model predicted. The second is a tendency to

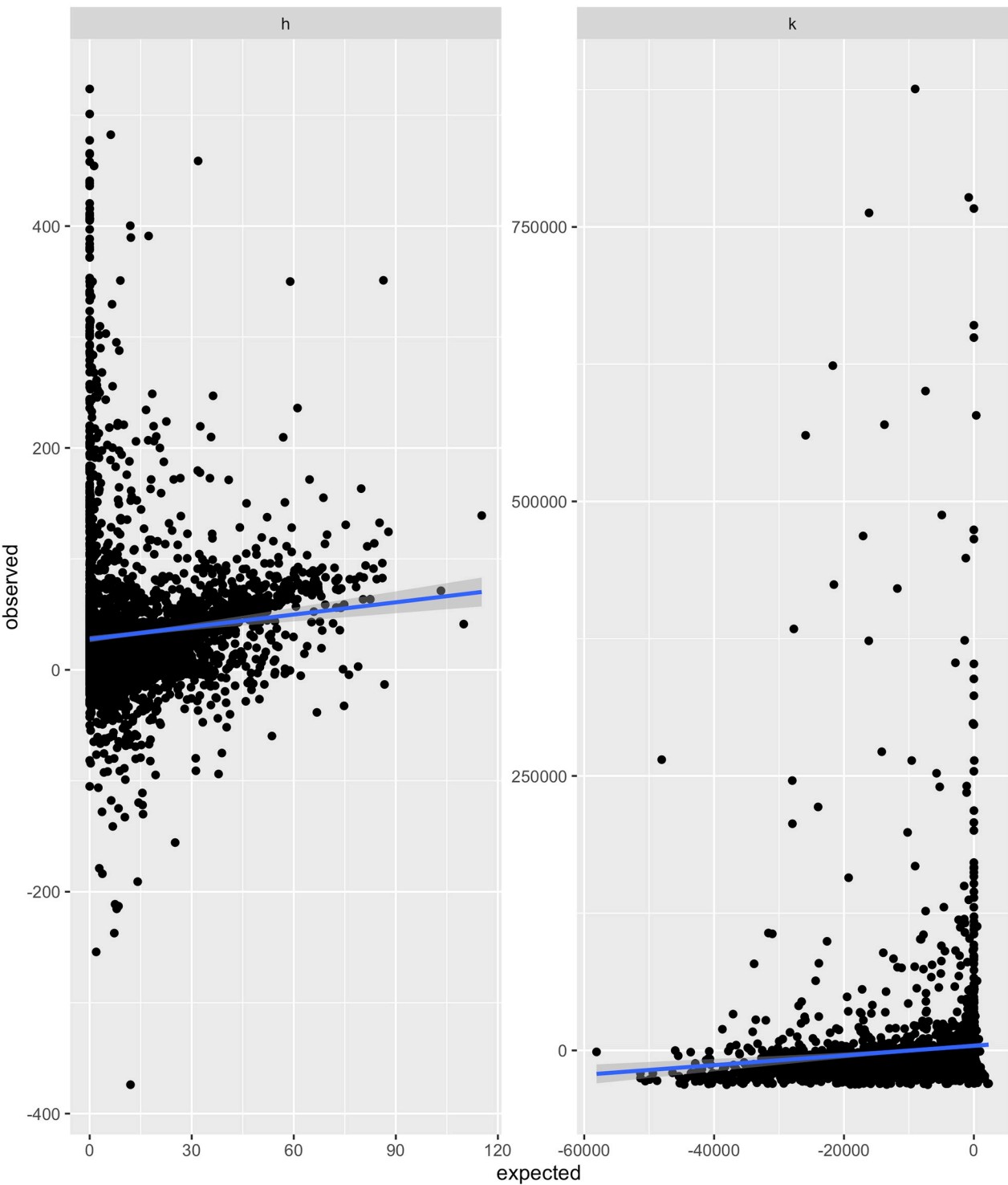

**Fig 1. Scatter plot of expected v.s. observed shift towards the red circle and the change in circle area relative to the blue circle.** (A) Shift towards the red circle. (B) Change in circle area.

shift and draw a larger circle when the model predicted no change. Likely, two phenomena are simultaneously occurring in this experiment, one that the Martins model can explain and the other the model cannot explain. We suspected that the two phenomena could be divided based on parameters determined in the game, and we developed this filter to separate the data

$$d < 0.8(r_{\text{blue}} + r_{\text{red}}). \tag{9}$$

Eq 9 separates the data based on whether the two circles shown to the participant were overlapping by 20% of their radiuses.

When we apply Eq 9 to the data, we find these results:

1. When the red and blue circles overlap and there is medium to high reliability, we can reject both hypotheses: (1) There is no correlation between the observed and expected shifts away from the blue circle; and (2) There is no correlation between the observed and expected change in circle area from the blue circle.

2. When the red and blue circles do not overlap, participants are making a choice to either stay with the blue circle, adopt the red circle or compromise 44% with the red circle when trust is high.

3. Participants tend to shift away from the red circle when trust is low, and the red and blue circles overlap.

## Data explained by the Martins model

Fig 2 is the result of applying Eq 9 to the data. In Fig 2A we can see that the filtered observations broadly match expectation with the linear model producing an $R^2 = 0.14$ (see Table 2). In Fig 2B the model performs noticeably worse with an $R^2 = 0.0029$ (see Table 2). The $p$-values for the slopes from the linear models are statistically significant compared to a Type I Error Rate of $\alpha = 0.05$ for the Medium and High trust scenarios this indicates that in these cases there is sufficient evidence to reject our null hypotheses and conclude that the observed results do concur with the extended Martins model predictions. In the Low trust scenarios, the $p$-values are not significant, indicating that in these scenarios, the extended Martins model results are not good predictors of the observed behaviour. In the Low trust scenarios individuals seem to move *away* from the red circle, which is counter to the assumptions of the extended Martins model. We elaborate more this negative shifting in its own section. Investigating Fig 2A reveals the effects of a confounding variable bounding observed values to a minimum value. We suspect this confounding variable to be the minimum size participants can draw their circle, which creates an artificial limit on circle size reduction.

We partitioned the data further by the reliability of the red circle, and Fig 3 shows such a partition. In general, the model predicts high trustworthiness interactions more accurately, and most of the outliers in the data lie within the low trustworthiness games. In particular Fig 3B, when $p = 0.2$ and $p = 0.5$, contain most of the unusually large data points compared with $p = 0.8$. We can explain this outlier behaviour as participants attempt unorthodox strategies to get the highest score since the red circle isn't a reliable source of information in those cases.

## Data unexplained by the Martins model

After investigating the data of the overlapping circles, we shifted focus to the data of the non-overlapping circles. Since the red and blue circles weren't overlapping in this case, participants would see two distinct circles. Therefore, we hypothesised that participants were choosing either to stick with the blue circle (what they know to be true), to 'take a leap of faith' and

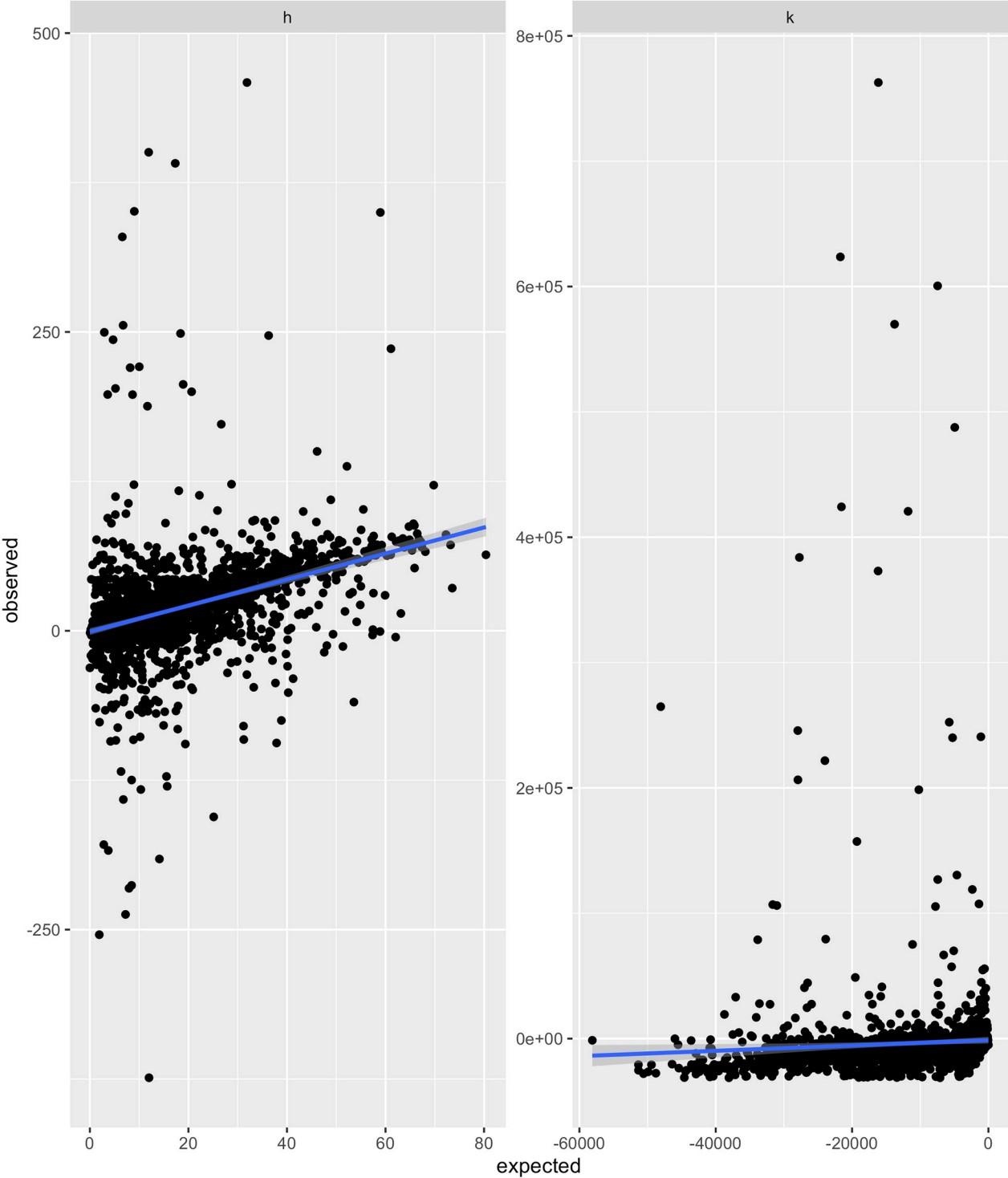

**Fig 2. Expected v.s. observed filtered based on games where the blue and red circle overlapped.** The solid red line is the line of best with intercept set to zero. (A) Shift towards the red circle relative to the blue circle. (B) Change in circle area relative to the blue circle.

**Table 2. The slope and $R^2$ values for Figs 2 and 3.**

| | Shift towards Red | | | | Change in circle area | | | |
|---|---|---|---|---|---|---|---|---|
| | Low | Mid | High | Total | Low | Mid | High | Total |
| slope | 0.17 | 0.94 | 1.14 | 1.06 | −0.02 | 0.23 | 1.08 | 0.27 |
| $p$-value | 0.58 | 1.06e−43 | 1.07e−114 | 1.07e−141 | 0.91 | 0.03 | 6.14e−39 | 1.16e−5 |
| $R^2$ | 0.01 | 0.09 | 0.15 | 0.14 | 0.01 | 0.01 | 0.05 | 0.003 |

The low, mid and high headings in the table refer to trust at low, medium and high values respectively, specifically they refer to parameter values $p = 0.2$, 0.5 and 0.8. Total is taking the data as a whole.

adopt the red circle as their new opinion, or to comprise with the red circle and draw their circle in between the red and blue circles. We tested this hunch by scaling the observed shift towards the red circle by the distance between the blue and red circles $d$ resulting in Fig 4. From Fig 4 we can see that the majority of the unpredicted shifts (73%) were between 0 and $d$ with most of these close to 0, which is consistent with the proposed explanation that the participants made a discrete choice between three options.

Similar to the data explained by the Martins model, we can separate the unexplained data based on trustworthiness which results in Fig 5. The tendency we expected to see, i.e. sticking with the known by drawing over the blue circle, is confined to the low trust scenarios of Fig 5, i.e. $p = 0.2$ and $p = 0.5$, but there is a tendency to comprise and a smaller chance to fully trust the red circle which contributes to a rightwards skew, particularly in the medium trust case of $p = 0.5$. The tendency to adopt or comprise with the red circle is unsurprisingly common in the high trust case of $p = 0.8$, and we can see distinguished peaks when $p = 0.8$, suggesting that participants are making a discrete choice concerning whether to fully, partially or not trust the red circle. Furthermore, we note the case when $p = 0.8$ relates closely to data collected by [14] when participants decided to comprise. In the study, [14] participants chose to adopt on average 40% of their interaction partner's opinion into their own, which is congruent with the average opinion shift in Table 3 of 0.44.

## Negative opinion shifting in the results

In the data, there has been a noticeable number of participants shifting away from the red circle, i.e. a negative opinion shift, which the Martins model does not predict. We have tabulated the number of negative opinion shifts in Table 4 and most negative shifts occur within five pixels of 0. A notable exception is when trust is low, i.e. $p = 0.2$, and the red and blue circles overlapped (explained data), but the majority of negative shifts that occurred were still below 50 pixels (10% the height of the play area and 14% the width) with only 10% of shifts being further than 50 pixels. Over both the explained and unexplained data sets, low trust games resulted in more negative shifts, whereas high trust games, i.e. $p = 0.8$, resulted in less negative shifts and both the explained and unexplained data produced similar proportional amounts of negative shifts. We can conclude that most of the negative opinion shifts occurring are from participants attempting to draw their circle on the blue circle, i.e. shifting by 0. In low trust scenarios, however, the negative shift could be more intentional, particularly for the explained data.

## Breakdown of individual participant involvement

To ascertain the influence of individual participants in the experimental data, we have developed Fig 6, which shows the number of games played versus the number of players. We uniquely identified participants through their IP addresses and used that information to count

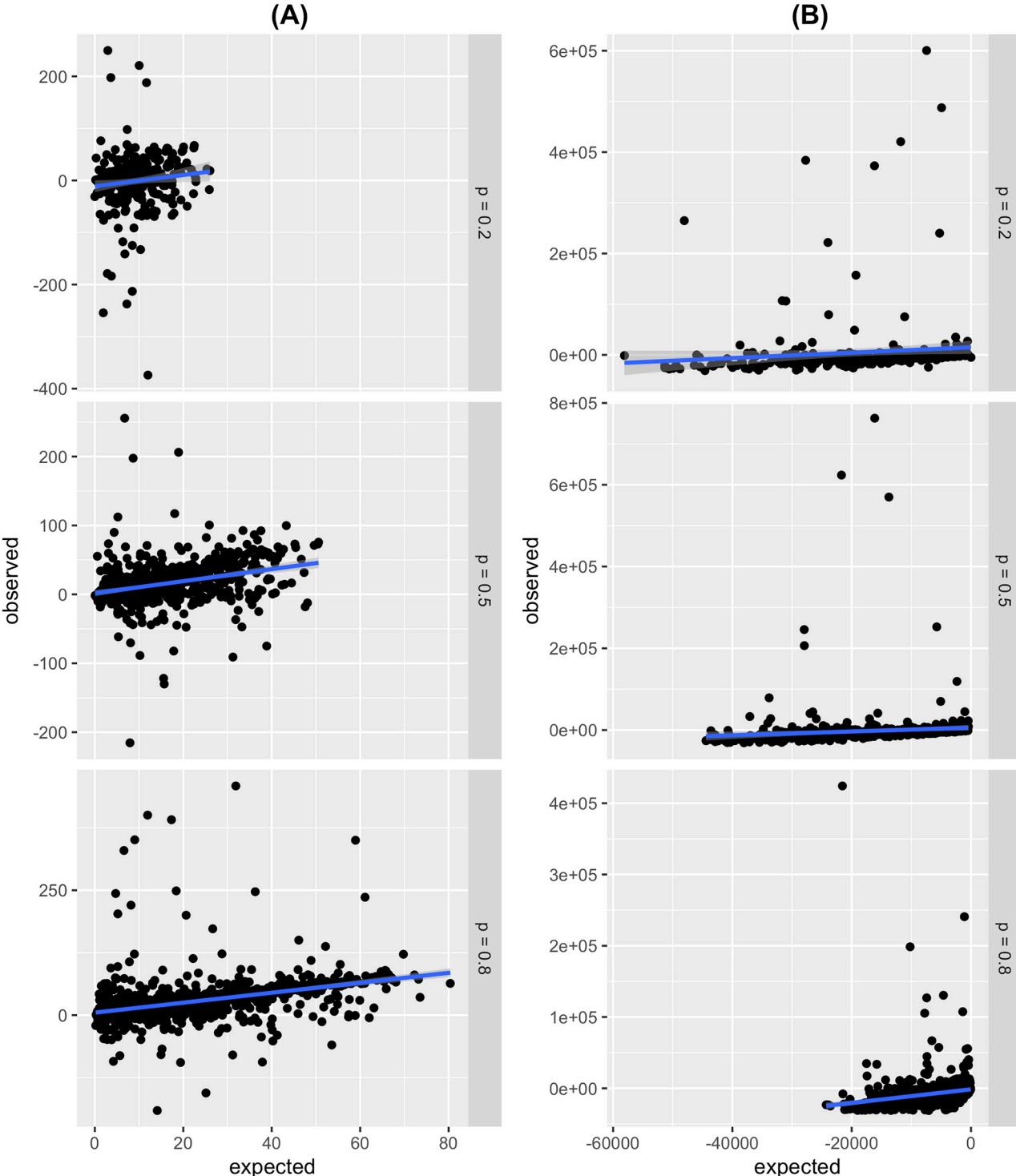

**Fig 3. Expected v.s. observed filtered based on games where the blue and red circle overlapped separated into different levels of trustworthiness.** The solid red line is the line of best with intercept set to zero. (A) Shift towards the red circle relative to the blue circle. (B) Change in circle area relative to the blue circle.

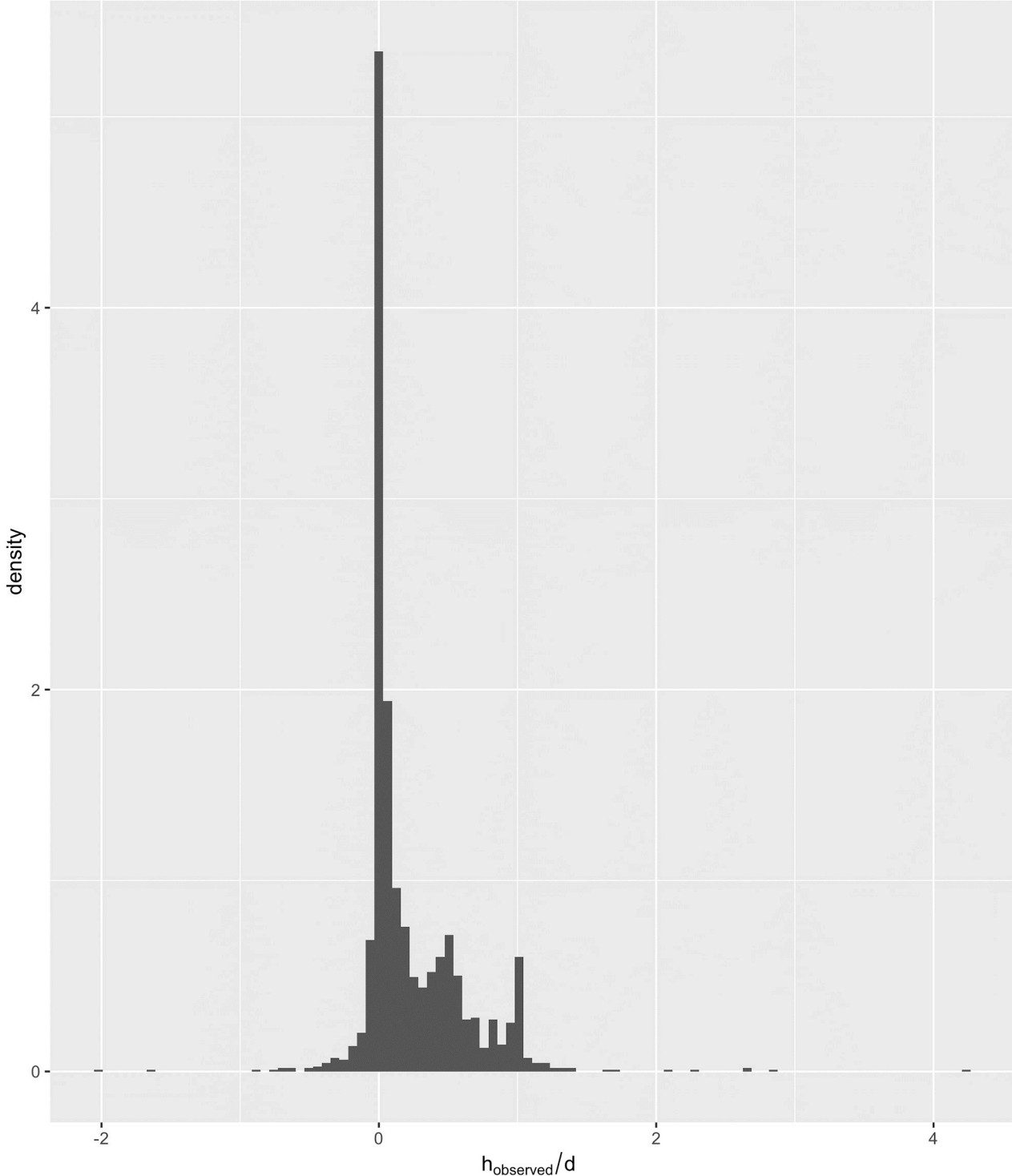

**Fig 4. Histogram of the shift towards the red circle relative to the total distance between the other and blue circle when the blue and red circles where not significantly overlapping.**

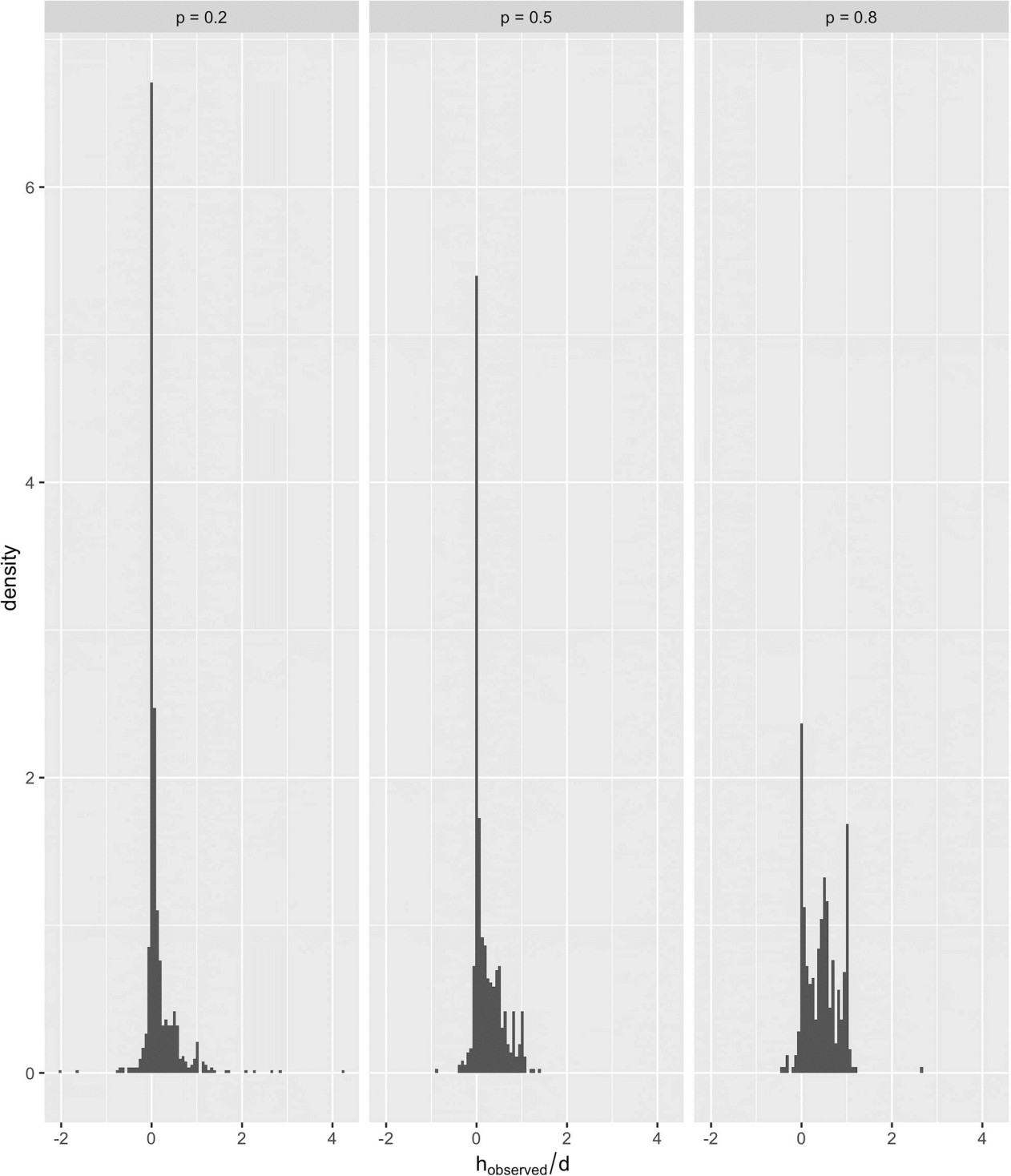

**Fig 5. Histogram of the shift towards the red circle relative to the total distance between the other and blue circle when the blue and red circles where not significantly overlapping separated into different levels of trustworthiness.**

**Table 3. The summary statistics for Figs 4 and 5 in pixels.**

|  | Low | Mid | High | Total |
|---|---|---|---|---|
| Mean | 0.13 | 0.22 | 0.44 | 0.23 |
| Median | 0.02 | 0.07 | 0.45 | 0.06 |
| Min | −2.05 | −0.91 | −0.47 | −2.05 |
| Max | 4.21 | 1.39 | 2.63 | 4.21 |
| 1st Quantile | −0.01 | 0.00 | 0.09 | 0.00 |
| 3rd Quantile | 0.15 | 0.39 | 0.72 | 0.42 |

The low, mid and high headings in the table refer to trust at low, medium and high values respectively, specifically they refer to parameter values $p$ = 0.2, 0.5 and 0.8. Total is taking the data as a whole.

**Table 4. Number of games that feature negative opinion shifting from the participants.**

| Observed Shifts (pixels) | Explained (games) | | | | Unexplained (games) | | | | Total (games) |
|---|---|---|---|---|---|---|---|---|---|
|  | Low | Mid | High | Total | Low | Mid | High | Total |  |
| $h_{observed} \geq 0$ | 163 | 454 | 895 | 1512 | 574 | 447 | 352 | 1373 | 2885 |
| $-5 \leq h_{observed} < 0$ | 36 | 58 | 69 | 163 | 141 | 68 | 26 | 235 | 398 |
| $-50 \leq h_{observed} < -5$ | 69 | 82 | 89 | 240 | 95 | 58 | 16 | 169 | 409 |
| $h_{observed} < -50$ | 31 | 9 | 11 | 51 | 14 | 3 | 0 | 17 | 68 |

The low, mid and high headings in the table refer to trust at low, medium and high values respectively, specifically they refer to parameter values $p$ = 0.2, 0.5 and 0.8. Total is the total games in a particular category, either explained, unexplained or across the whole experiment.

how many games a participant played. The median games played was 10 games whereas the mean games played was 14.63 games this suggests there is a skew in the histogram Fig 6. Furthermore, the top 10% of participants (in terms of games played) are responsible for only 35% of all games played in the experiment and 80% of participants played 23 or fewer games. The high number of games played by a small number of players presents a potential issue because their "learning" could bias the results if we assume that games are independent trials for the purposes of analysis. That is, if players' scores improve as they played additional games, the independence assumption would be invalid, tainting the results of our analyses. To investigate whether participants were learning to play the game, we compiled Table 5 showing the mean scores for the $n$th game played. While for any number of attempts there is a broad range of scores, including at the extremes, we can see that average participants' score doesn't increase as they play more games suggesting that participants are not "learning" to play the game better or at least not learning to improve their scores.

## Discussion

The data we have collected has produced surprising and interesting results. We have identified two different types of behaviour in the experiment. The first type of behaviour is congruent with the predictions made by the Martins model, while the second fell outside the scope of the Martins model, and we were able to distinguish between the two behaviours by developing Eq 9, which divides the data based on the red and blue circle overlap. When investigating the second dataset, we developed Figs 4 and 5, and we concluded that participants are treating the problem of finding the dot as a discrete choice, i.e. it must be in either the red circle or the blue. Adopting the red circle opinion is contrary to the Martins model, which considers distant

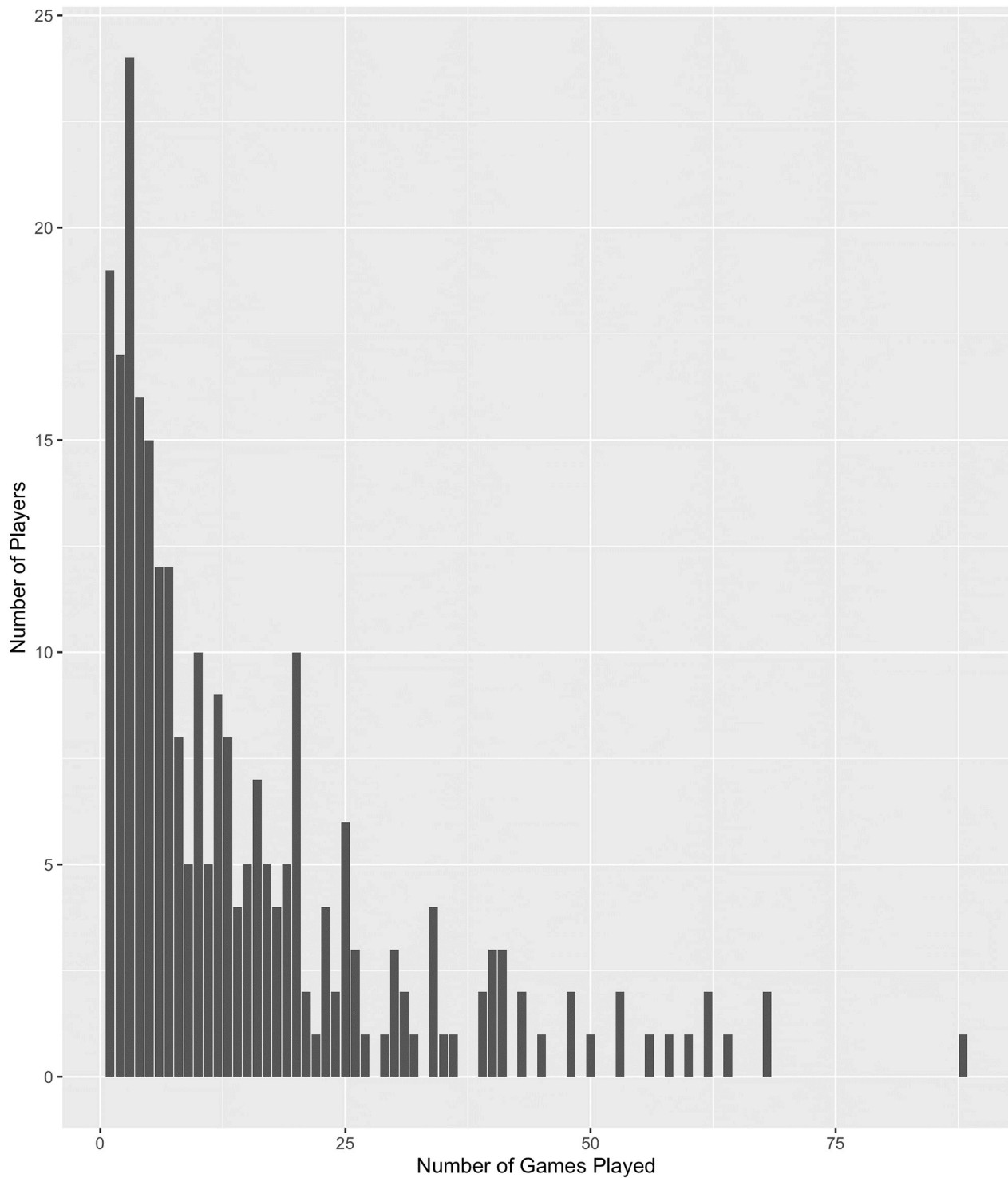

**Fig 6. Histogram of the participant frequency on the number of games played.**

**Table 5. The mean score of participants by attempts.**

| Attempt | Mean Score |
|---|---|
| 1st | 3230 |
| 2nd | 1983 |
| 3rd | 4896 |
| 4th | 2533 |
| 5th | 3563 |
| 6th | 5072 |
| 7th | 3195 |
| 8th | 2028 |
| 9th | 4358 |
| 10th | 3690 |
| 11th-15th | 4148 |
| 16th-20th | 4039 |
| 21st-25th | 4212 |
| 26th-30th | 5124 |
| 31st-40th | 3136 |
| 41st-50th | 3007 |
| 51st-88th | 3909 |

opinions as "untrustworthy" even with a *p* close to 1. Thus, according to the Martins model, a participant should always ignore the red circle. There are multiple causes for participants to switch to a discrete choice mindset. It is either the result of cognitive bias to simplify the problem or devised from the instructions we gave to participants. The instructions explained that M. Dotte is the one who reveals the blue circle. Despite the instructions explaining that M. Dotte is always reliable, participants might still doubt M. Dotte and not fully internalise the blue circle as their opinion. Although when we compare the unexplained results to the results in the literature, we see startling agreement. We observed in Fig 5 that when reliability is low, participants tended to keep their opinions close to the blue circle, shifting towards the red circle at an average of 0.2*d*. But with increased reliability, participants began to "compromise" with the red circle by drawing their circle at 0.4*d* from the blue to the red circle. This 0.4 magnitude shift agrees closely with the results in [14], when participants decided to compromise.

The Martins model appears ineffective in predicting the change in the circle area. We note in Fig 2B that a confounding variable is bounding the observed change in the circle area, thus forcing a minimum value. We posit that the confounding variable is the minimum circle size (a five-pixel radius) relative to the size of the play area, i.e. the box. We can see a linear tread exists in Fig 2B and is cut by the minimum circle size boundary. The outliers are more extreme in Fig 2B than for Fig 2A, and we can surmise that the outliers for Fig 2B were participants' attempts to 'break' the game. Essentially the participants were testing if drawing a large circle would net a substantial number of points. When the data is separated based on reliability, it is clear that the Martins model predicts high-reliability scenarios for circle area change more accurately. In that case, the prediction for circle area change is small enough so that participants can draw circles of those sizes, thus above the minimum circle size boundary.

There is much debate over whether opinions can be "negatively" influenced, i.e. when individuals' opinion difference is large, the distance between the individuals' opinions increases after an interaction (i.e. negative opinion shift). Negative opinion influence is not to be confused with a negative opinion shift, which is an unconditional shift away from an interaction partner's opinion, i.e. negative opinion shifts may occur without negative influence. Some

theoretical opinion dynamics models [18–20] rely on negative opinion influence to create polarisation, and others like [21] predict negative opinion shift resulting from negative opinion influence (but not necessarily resulting polarisation) in a discrete opinion context. In contrast, empirical experiments that attempted to measure negative opinion influence have so far failed. For example, [22], although finding evidence of negative opinion shifts, found no evidence of negative opinion influence.

The data we collected conforms with the results in [22], we observed in our data negative opinion shifts, but it was localised to when the red and blue circles were overlapping, not when the circles were distant. Most negative shifts resulted from participants attempting to draw onto the blue circle and were within 5 pixels of the blue circle. Only when reliability was low and the red and blue circles overlapped did it appear like participants intentionally shifted away from the red circle. The Martins model only predicts positive opinion shifts and thus does not explain the negative opinion shifting occurring at low reliability. The negative opin-ion shifting is likely the phenomenon which causes the Martins model to be a poor predictor of low-reliability interactions. We theorise that participants, when presented with a low-reli-ability red circle close to the blue circle, believe that the red circle reduces the chances that the dot is in the blue circle and thus moves away from the red, which the Martins model does not consider. Although, we see similar behaviour in the model developed in [21].

## Conclusion

In this paper, we aimed to verify whether the Martins model is an accurate model of opinion exchange. We can conclude from the data that the Martins model is only accurate in specific circumstances. Specifically, we can reject the two null hypotheses when the red and blue circles overlap for medium to high reliability. Furthermore, we identified two phenomena occurring in this experiment; along with the phenomena explained by the Martins model, we observed participants making discrete choices. The discrete choice behaviour exclusively occurred when the red and blue circles didn't significantly overlap. We conjectured that the discrete choice occurred due to the human need to simplify the problem or participants not completely trust-ing the blue circle as their own opinion. Either way, this highlights the multifaceted nature of opinion exchange and illustrates the context-sensitivity of human behaviour. For a model of opinion exchange to sufficiently capture the complexities of interactions, the model would need to navigate the context of an interaction. Essentially the model needs to switch between discrete and continuous opinions when appropriate creating a complete synthesis of a discrete and continuous opinion model.

The Martins model predicted the opinion shifts of participants with reasonable precision when only considering the data explained by the Martins model. The $R^2$ for the linear tread lines are low because of the presence of outliers and a significant number of interactions that resulted in negative opinion shift, but from Fig 2 it is clear that there exists a linear trend. We can conclude that the Martins model predicts the general behaviour of the participants when there is a significant overlap between the red and blue circles. This conclusion is weak, and a more robust experiment with more participants is needed to determine whether the Martins model predicts human behaviour. Due to the simplicity of our experiment design, it should be easy to recreate this experiment at scale.

## Supporting information

**S1 Fig. The instruction given to every participant.**
(TIF)

**S2 Fig. An example of the game a participant would of played when a participant is exposed to the blue circle.**
(TIF)

**S3 Fig. An example of the game a participant would of played when a participant is exposed to the red circle.**
(TIF)

**S4 Fig. An example of a completed the game for a participant.**
(TIF)

**S1 File. The experiments website source code.**
(ZIP)

**S2 File. Raw data collected from the experiment.**
(CSV)

**S1 Appendix. Equations of the Martins model.**
(PDF)

## Acknowledgments

We would like to acknowledge the contribution from the QUT eResearch team: Ryan Bennett, Mitchell Haring, Yvette Wyborn, Craig Windell and Adam Smith, for their contribution in setting up, running and testing the website involved in this study.

## Author Contributions

**Conceptualization:** Johnathan A. Adams, Gentry White, Robyn P. Araujo.

**Data curation:** Johnathan A. Adams, Gentry White.

**Formal analysis:** Johnathan A. Adams.

**Investigation:** Johnathan A. Adams.

**Methodology:** Johnathan A. Adams.

**Software:** Johnathan A. Adams.

**Supervision:** Gentry White, Robyn P. Araujo.

**Visualization:** Johnathan A. Adams.

**Writing – original draft:** Johnathan A. Adams.

**Writing – review & editing:** Johnathan A. Adams, Gentry White, Robyn P. Araujo.

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
