## [Decision Letter · Decision Letter 0]

13 May 2022

PONE-D-22-10361Person-to-person opinion dynamics: an empirical study using an online gamePLOS ONE

Dear Dr. Adams,

Thank you for submitting your manuscript to PLOS ONE. After careful consideration, we feel that it has merit but does not fully meet PLOS ONE’s publication criteria as it currently stands. Therefore, we invite you to submit a revised version of the manuscript that addresses the points raised during the review process.

Both reviewers find the work meritorious and of interest after a serious and thorough analysis of the work. Notwithstanding some of the objections and possible biases in the design of experiments, the explanations of the other elements pointed out by the reviewers deserve a detailed response before considering the work for potential publication. I look forward to a detailed review of the paper.

We look forward to receiving your revised manuscript.

Kind regards,

José Manuel Galán, Ph.D.

Academic Editor

PLOS ONE

Journal Requirements:

"Robyn P. Araujo is the recipient of an Australian Research Council (ARC) Future Fellowship (project number FT190100645) funded by the Australian Government"

"Robyn P. Araujo is the recipient of an Australian Research Council (ARC) (https://www.arc.gov.au/) Future Fellowship (project number FT190100645) funded by the Australian Government.

Additional Editor Comments:

Both reviewers find the work meritorious and of interest after a serious and thorough analysis of the work. Notwithstanding some of the objections and possible biases in the design of experiments, the explanations of the other elements pointed out by the reviewers deserve a detailed response before considering the work for potential publication. I look forward to a detailed review of the paper.

Reviewers' comments:

Reviewer's Responses to Questions

**Comments to the Author**

1. Is the manuscript technically sound, and do the data support the conclusions?

Reviewer #1: Partly

Reviewer #2: Partly

2. Has the statistical analysis been performed appropriately and rigorously? 

Reviewer #1: Yes

Reviewer #2: Yes

3. Have the authors made all data underlying the findings in their manuscript fully available?

Reviewer #1: Yes

Reviewer #2: Yes

4. Is the manuscript presented in an intelligible fashion and written in standard English?

Reviewer #1: Yes

Reviewer #2: Yes

5. Review Comments to the Author

Reviewer #1: I found the experiment carried out by the authors very interesting to verify opinion models, especially the more abstract approach they have shown in the experiment compared to previous works. I consider this work interesting and innovative enough to be published in this journal. However, below I have proposed some minor changes that I think could improve the presented paper:

- The authors do a very extensive literature review of previous experiments, explaining them in great detail, but I miss much more information about their own. Nowhere is it explained what instructions are given to the participants about how the experiment works, how the scores given to them as a result of a correct guess are calculated or what kind of reward the participants receive in the experiment. I think that all this information should be included in order to better understand the behaviour of the participants, as they can have a great influence on the conclusions. For example, in the discussion section, the authors conclude that “participants are treating the problem of finding the dot as a discret choice, i.e. it must be in either in the red circle or the blue”. This behaviour may be a consequence of the game's instructions not being clear enough.

- The authors explain that the participants were encouraged to play again and that in total 3760 games were played by a total of 257 unique users. It would be very interesting to know the distribution of the total number of games played by each participant, to check if the results could be influenced by someone who has repeated the game too many times. In addition, an analysis could be made of whether there exists a "learning curve" in the participants, i.e. if the score obtained by each participant improves as the number of games increases. This would indicate that the observations are not independent, and therefore would need to be taken into account.

- I suggest the authors to change the name of the variable Δx confusing. This variable represents not only a variation in x, but in the distance between the centres of the two circles. I think another name would be more appropriate.

- The condition (7) presented by the authors does not represent whether the two circles overlap. The correct condition would be Δx < r_blue + r_red. This condition should be corrected or the reason for the choice should be explained.

- I suggest the authors to improve the quality of the images included in the manuscript, as the it is very low. In figure 3d it is not even possible to distinguish correctly the red line and in figure 5 the histograms are not visible because they are partially covered by the legend.

- I found some typos while reading the text (e.g. "radii" just after equation 1 or "Of the data that the Matins model" in the conclusions section). I encourage the authors to read all the text again and correct them.

Reviewer #2: Referee Report – Manuscript PONE-D-22-10361

“Person-to-person opinion dynamics: an empirical study using an online game”

In this work, authors carry out an experiment trying to emulate the Martins model on opinion dynamics. In their experiment, participants must guess the location of a hidden dot in a space. They are given two pieces of information: two circles representing the possible location of the hidden object. Participants first observe a blue circle with 80% accuracy level and then a red circle with either a 20%, 50% or 80% accuracy level. They are finally asked to indicate where they believe the dot is, also by representing a circle. Authors find that the Martins model only explains part of the observed results. In particular, the behaviour when there is a significant overlapping of the blue and red circles. When there is no overlapping, the prediction ability of the model is lower. Authors indicate that the possible explanation of this is that participants may be treating the game as a discrete choice between the red and the blue circles rather than as a continuous choice in all the space.

The paper is novel in its objective of experimentally testing the Martins model, but I have some concerns, especially regarding the experimental design. In what follows, I will present these concerns, along with some recommendations which I hope can help authors improve the quality of their paper.

MAJOR COMMENTS

Introduction:

1. I have missed a summary of the experimental design and a summary of the main results in the introduction.

Previous experiments:

1. I like the way in which you present these previous works but would like to know the relevance of some papers you only mention but do not explain, like references [9], [10] and [12].

2. Furthermore, what is the contribution of your work to these previous papers?

Materials and methods:

1. Where participants incentivized somehow as to elicit truthful behaviour?

2. How did you treat the data of participants who left in the middle of the game? Was this frequent?

The experiment:

1. My main concern with this work comes in this regard. As the game is designed it resembles more a situation of evaluating a perfect signal (blue circle) vs. an imperfect signal (red circle) rather than personal vs. outside information. I know that the blue circle is not perfect, but it gives the highest possible level of accuracy they can have (80%). I personally find it challenging to interpret that the blue circle represents an individual’s initial opinion and that the red circle represents the opinion of another individual. I see it as two signals of different quality whereas opinions have deeper connotations. For instance, if the red circle comes with a green light (80% accuracy), there should be some kind of bias towards the blue circle as it represents one’s opinion. When designing an experiment, it is difficult to find the equilibrium between making something abstract as to minimise certain biases, but at the same time, appliable to the real world. In this case, I find it particularly difficult to transfer this setting to opinion dynamics if the blue circle does not count with a more “personal touch”, let’s say.

2. Be careful with the use of red and blue in the game, as experimental evidence has shown that these colours may carry some political (and mostly unconscious) connotations. You could do a small control with the colours reversed as to discard any colour effect.

3. Be also careful with the order in which participants receive the two pieces of information. There could be an order effect, specially when the red circle comes with a green light, and one of the pieces of information has more relevance because of the order in which this information is shown. I recommend you to check literature about information disclosure in this line. If there is evidence that we usually pay more attention to the first piece of information than to the second one, this could capture the previous idea of the bias towards the blue circle (one’s opinion), that I pointed out before. Otherwise, you could also do a small control where you show the circles in the reversed order and check that your results are robust to the order in which information is disclosed.

4. If I understood correctly, participants could play as many times as possible. If this is the case, I am concerned about how you treat this data. Repeating the game allows for a learning process but if they can repeat it as many times as they wish, you are allowing for different learning degrees. One could leave it after just one try and another one could play 20 times. Are you treating the first’s unique attempt as the 20th attempt of the second participant or are you distinguishing them somehow?

Model and Data Predictions:

1. Some intuition about the data predictions could help the reader follow this section and prepare him/her for the results.

2. From my point of view, presenting a set of hypotheses to be later tested would also be useful for the reader.

Results:

1. I personally found this section specially challenging to follow. It is the most important part of the paper and I believe it deserves some special attention. Please, reconsider the exposition of your results in this section. Some paragraphs seem repetitive and other results may go unnoticed. I suggest you to enumerate your results as to make them clearer for the reader.

Discussion and conclusion:

1. These sections clarify the results section. Beyond finding to what extent the Martins model explains these experimental results, I would appreciate if you could also translate this to the topic at hand: opinions dynamic. What do these results indicate us about how we exchange opinions in the real world?

2. Are there any other interpretations of why the Martins model fails to explain behaviour when circles do not overlap? Behavioural biases? Participants not correctly understanding what the best outcome for them was during the experiment?

MINOR COMMENTS

1. Please revise typos in lines 71, 84, 186, 218, 237, 248, 253, 254, 255 and 288.

2. I suggest you to read the following paper and related works, just in case they seem relevant in the experimental literature on opinions dynamics:

- Battiston, P., & Stanca, L. (2015). Boundedly rational opinion dynamics in social networks: Does indegree matter?. Journal of Economic Behavior & Organization, 119, 400-421.

6. PLOS authors have the option to publish the peer review history of their article (what does this mean?). If published, this will include your full peer review and any attached files.

Reviewer #1: **Yes: **Diego Escribano Gómez

Reviewer #2: No

---

## [Author Response · Author response to Decision Letter 0]

11 Jul 2022

Reviewer #1: 

- The authors do a very extensive literature review of previous experiments, explaining them in great detail, but I miss much more information about their own. Nowhere is it explained what instructions are given to the participants about how the experiment works, how the scores given to them as a result of a correct guess are calculated or what kind of reward the participants receive in the experiment. I think that all this information should be included in order to better understand the behaviour of the participants, as they can have a great influence on the conclusions. For example, in the discussion section, the authors conclude that “participants are treating the problem of finding the dot as a discrete choice, i.e. it must be in either in the red circle or the blue”. This behaviour may be a consequence of the game's instructions not being clear enough.

Response: We apologise for the unclear instructions and insufficient detail on the game and scoring. We have addressed this by amending the Experiments section, including a subsection discussing the instructions given to the participants in detail. We have also included a subsection focusing on scoring to engage participants and encourage further play. We note that the reviewer’s concerns about lack of clarity in the instructions to the player are interesting but would note that the instructions in the actual game were much more explicit than those initially described in the paper; these instructions are now included explicitly in the revised manuscript. 

- The authors explain that the participants were encouraged to play again and that in total 3760 games were played by a total of 257 unique users. It would be very interesting to know the distribution of the total number of games played by each participant, to check if the results could be influenced by someone who has repeated the game too many times. In addition, an analysis could be made of whether there exists a "learning curve" in the participants, i.e. if the score obtained by each participant improves as the number of games increases. This would indicate that the observations are not independent, and therefore would need to be taken into account.

Response: We have now included a new subsection in Results investigating this issue in detail. We would like to thank both reviewers for giving us the opportunity to include this important analysis in the paper. Results show that there is no significant learning over multiple plays.

- I suggest the authors to change the name of the variable Δx confusing. This variable represents not only a variation in x, but in the distance between the centres of the two circles. I think another name would be more appropriate.

Response: We have changed Δx to dx to be clearer and avoid confusion. 

- The condition (7) presented by the authors does not represent whether the two circles overlap. The correct condition would be Δx < r_blue + r_red. This condition should be corrected or the reason for the choice should be explained.

Response: We apologise for the lack of clarity in the notation and have re-written the condition as suggested. 

- I suggest the authors to improve the quality of the images included in the manuscript, as the it is very low. In figure 3d it is not even possible to distinguish correctly the red line and in figure 5 the histograms are not visible because they are partially covered by the legend.

Response: We have reviewed and re-produced the figures for improved clarity and readability, 

- I found some typos while reading the text (e.g. "radii" just after equation 1 or "Of the data that the Matins model" in the conclusions section). I encourage the authors to read all the text again and correct them.

Response: We apologise for the oversight in proofing the manuscript and have thoroughly reviewed the manuscript correcting the typographical errors. 

Reviewer #2: Referee Report – Manuscript PONE-D-22-10361

“Person-to-person opinion dynamics: an empirical study using an online game”

MAJOR COMMENTS

Introduction:

1. I have missed a summary of the experimental design and a summary of the main results in the introduction.

Response: We apologise for the omission and have now included a summary of the experimental design and results in the Introduction.

Previous experiments:

1. I like the way in which you present these previous works but would like to know the relevance of some papers you only mention but do not explain, like references [9], [10] and [12].

2. Furthermore, what is the contribution of your work to these previous papers?

Response: We have now expanded on the literature review section to contextualise the cited papers and explain their relevance to our paper, and well as the context of our paper with respect to the broader literature. 

Materials and methods:

1. Where participants incentivized somehow as to elicit truthful behaviour?

Response: Participants were not materially incentivised; this was made clear to the participants with a prominent disclaimer on the website. The only incentive is participants’ personal desire to maximise their score in each game. The score was framed in ‘$’, which is part of the game’s backstory designed to engage players and get them personally invested in the game’s outcome.

2. How did you treat the data of participants who left in the middle of the game? Was this frequent?

Response: If a participant left in the middle game we recorded no data, thus have no record of incomplete games. This is now explicitly stated in the manuscript. 

The experiment:

1. My main concern with this work comes in this regard. As the game is designed it resembles more a situation of evaluating a perfect signal (blue circle) vs. an imperfect signal (red circle) rather than personal vs. outside information. I know that the blue circle is not perfect, but it gives the highest possible level of accuracy they can have (80%). I personally find it challenging to interpret that the blue circle represents an individual’s initial opinion and that the red circle represents the opinion of another individual. I see it as two signals of different quality whereas opinions have deeper connotations. For instance, if the red circle comes with a green light (80% accuracy), there should be some kind of bias towards the blue circle as it represents one’s opinion. When designing an experiment, it is difficult to find the equilibrium between making something abstract as to minimise certain biases, but at the same time, appliable to the real world. In this case, I find it particularly difficult to transfer this setting to opinion dynamics if the blue circle does not count with a more “personal touch”, let’s say.

Response: We appreciate the careful and considered thought the reviewer gave our paper. We should clarify that the score given the red circle is not a measure of accuracy but is a measure of the probability that the circle contains any information, i.e. its reliability as a source of information about the location of the dot. The blue circle is the initial piece of information given to the player and is 100% reliable; hence, the player will likely adopt this as "their" belief about the location of the dot. The game's instructions, which we have now included in the revised manuscript, clearly explained this to the players. 

Evaluating the red circle is largely a question of trust, i.e. "do I trust that the red circle is giving me useful information?" Individuals' bias towards the blue circle is measured as h_observed and compared to the value predicted by the modified Martin's model h_expected, which accounts for the reliability of the red circle. 

We can further agree with the reviewer's comments regarding experimental design challenges, particularly in opinion dynamics. We believe that the backstory given to the players (now included in the new S1 Fig) along with the structure of the game would engage players and, by design, encourage them to adopt the blue circle as their own opinion, thus leading to a more personal investment in the game. 

Player deference to the blue circle as their opinion is borne out by the congruence between player behaviour and the behaviour predicted by the modified Martins model (which assumes that the blue circle is the player's initial opinion). Because the game is an abstract (context free) exercise, we do not need to take the initial steps seen in other experiments (eg. [ref]) of measuring individuals initial or baseline opinions. Instead, we supply this in the form of the blue circle. Asking general knowledge questions, like in Mossilid 2013, or asking for stances on political issues, relies on measuring those initial opinions/answers, which may introduce external biases. Our abstract approach improves upon this issue by directly controlling a participant's initial opinion.

2. Be careful with the use of red and blue in the game, as experimental evidence has shown that these colours may carry some political (and mostly unconscious) connotations. You could do a small control with the colours reversed as to discard any colour effect.

Response: We agree that this presents a potential issue for the manuscript. We believe that the manuscript avoids political bias from the choice of colours because the experiment is abstract enough to avoid any political association. In addition, the participants are sourced online, thereby drawing on an international population, and political parties sharing common colours across different countries may hold very different political viewpoints.

We chose the colours blue and red for accessibility (i.e. using websafe high-contrast colours and avoiding issues with dichromatics). Political colour bias might contribute to the noise in the data, but we believe that individual colour biases should effectively cancel each other out. Geotagging the IP addresses of the participants to break down the data by country would have violated our ethics approval; thus, it is impossible to determine the location of individuals and any potential political colour bias. We have included additional discussion of these points in our revised discussion section.

3. Be also careful with the order in which participants receive the two pieces of information. There could be an order effect, specially when the red circle comes with a green light, and one of the pieces of information has more relevance because of the order in which this information is shown. I recommend you to check literature about information disclosure in this line. If there is evidence that we usually pay more attention to the first piece of information than to the second one, this could capture the previous idea of the bias towards the blue circle (one’s opinion), that I pointed out before. Otherwise, you could also do a small control where you show the circles in the reversed order and check that your results are robust to the order in which information is disclosed.

Response: We respectfully emphasize that the blue circle represents the player’s own initial opinion, hence we show the blue circle to the participants first to take advantage of any order bias and cement the blue circle in the participants’ minds as “their” opinion. As we clarified above, the experiment is not designed to measure participants’ interpretation of perfect vs imperfect signals (in which case, randomising the order in which participants are presented with information would have been essential). 

4. If I understood correctly, participants could play as many times as possible. If this is the case, I am concerned about how you treat this data. Repeating the game allows for a learning process but if they can repeat it as many times as they wish, you are allowing for different learning degrees. One could leave it after just one try and another one could play 20 times. Are you treating the first’s unique attempt as the 20th attempt of the second participant or are you distinguishing them somehow?

Response: The reviewer makes an excellent point. We have now added two different analyses to our revised manuscript to address this issue. Specifically, (1) we calculated the average score obtained on every attempt. We found that the average score of the first attempt is not substantially nor significantly different from the average score of subsequent attempts (see Table 5 in revised manuscript), suggesting that no learning is occurring and that participants behave the same regardless of the attempt. We therefore consider that treating the first attempt of a participant equally to later attempts is justified. (2) As we now highlight in our revised manuscript, 80% people played up to 23 game.

Model and Data Predictions:

1. Some intuition about the data predictions could help the reader follow this section and prepare him/her for the results.

2. From my point of view, presenting a set of hypotheses to be later tested would also be useful for the reader.

Response: These are excellent suggestions, and we have now updated our manuscript to include a hypothesis at the end of Model and Data Predictions while also elaborating on the intuition of the predictions. 

Results:

1. I personally found this section specially challenging to follow. It is the most important part of the paper and I believe it deserves some special attention. Please, reconsider the exposition of your results in this section. Some paragraphs seem repetitive and other results may go unnoticed. I suggest you to enumerate your results as to make them clearer for the reader.

Response: We appreciate this thoughtful suggestion. We have now carefully revised our Results section, and segmented the our results into sub-sections. We also add clarity by enumerating our overarching findings in the introduction to our Results section.

Discussion and conclusion:

1. These sections clarify the results section. Beyond finding to what extent the Martins model explains these experimental results, I would appreciate if you could also translate this to the topic at hand: opinions dynamic. What do these results indicate us about how we exchange opinions in the real world?

Response: Polarisation in the Martins model is caused by mistrust, which is encapsulated by the parameter p, i.e. the probability that an individual will share misinformation. However, p does not directly influence an agent's new opinion when interacting, but through an intermediate variable p* - the result of applying Bayes' theorem to the model. The variable p* modifies p to reflect how more or less trustworthy an individual is. For instance, if an agent interacts with someone of very different opinion, then, according to the Martins model, the agent will believe the other agent less because, through Bayesian inference, the first agent deduces that the other agent is likely to be incorrect. The Martins model explains confirmation bias and polarisation as the result of individuals understanding that misinformation exists, and using that knowledge to reject new and different information because, based on their own opinion, the new and different information is more likely to be misinformation. There are two aspects to our results that illuminate the Martins model. In the first instance when interacting agents have relatively similar opinions, the Martins model predicts their behaviour well. In the second instance when the interacting agents have drastically differing opinions the Martins model is a poor predictor of the outcome of their interaction. The Martins model expects no shift to occur when the circle do not overlap since that would be clear evidence that the red circle is misinformed. Instead, we see participants compromising or adopting the red opinion, suggesting that when the circles are distinct participants start thinking in a discrete opinion context. The Martins model considers opinions as continuous thus the model is insufficient to describe these situations. When there is overlap between the circles, the model is surprisingly accurate in predicting opinion shift, suggesting that the model's explanation of confirmation bias is justified.

2. Are there any other interpretations of why the Martins model fails to explain behaviour when circles do not overlap? Behavioural biases? Participants not correctly understanding what the best outcome for them was during the experiment?

Response: There are multiple possible reasons why the Martins model fails to predict the behaviour, but the simplest explanation is that participants treat the game as a discrete choice problem. We can build on this explanation by speculating that the switch to discrete thinking is motivated by a cognitive bias to simplify the problem. If the participants did not understand the best outcome, we would expect no distinguished peaks in Figs 4 and 5, yet we do see distinguishable peaks. Participants understood that to maximise their score, they needed to guess better than the blue circle and attempted to incorporate the limited information provided by the red circle.

MINOR COMMENTS

1. Please revise typos in lines 71, 84, 186, 218, 237, 248, 253, 254, 255 and 288.

Response: We appreciate the reviewer’s consideration and conducted a thorough proofreading of the manuscript, correcting all typos.

2. I suggest you to read the following paper and related works, just in case they seem relevant in the experimental literature on opinions dynamics:

- Battiston, P., & Stanca, L. (2015). Boundedly rational opinion dynamics in social networks: Does indegree matter?. Journal of Economic Behavior & Organization, 119, 400-421.

Response: We thank the reviewer for the suggestion (and complete citation) of the paper by Battison and Stanca. Although their paper, which focuses on behaviour on social networks, is not directly related to our work, we found numerous authors' insights and comments useful. We now cite this work in our updated literature review.

---

## [Decision Letter · Decision Letter 1]

19 Sep 2022

Person-to-person opinion dynamics: an empirical study using an online game

PONE-D-22-10361R1

Dear Dr. Adams,

We’re pleased to inform you that your manuscript has been judged scientifically suitable for publication and will be formally accepted for publication once it meets all outstanding technical requirements.

Kind regards,

José Manuel Galán, Ph.D.

Academic Editor

PLOS ONE

Additional Editor Comments (optional):

Both reviewers consider that most of the issues raised in the original submission have been resolved. Reviewer 1 requests clarification of some minor aspects to make the article easier to understand. Please try to address those suggestions in your final submission, but I consider the paper worthy of publication.

Reviewers' comments:

Reviewer's Responses to Questions

**Comments to the Author**

1. If the authors have adequately addressed your comments raised in a previous round of review and you feel that this manuscript is now acceptable for publication, you may indicate that here to bypass the “Comments to the Author” section, enter your conflict of interest statement in the “Confidential to Editor” section, and submit your "Accept" recommendation.

Reviewer #1: All comments have been addressed

Reviewer #2: All comments have been addressed

2. Is the manuscript technically sound, and do the data support the conclusions?

Reviewer #1: Yes

Reviewer #2: Yes

3. Has the statistical analysis been performed appropriately and rigorously? 

Reviewer #1: Yes

Reviewer #2: Yes

4. Have the authors made all data underlying the findings in their manuscript fully available?

Reviewer #1: Yes

Reviewer #2: Yes

5. Is the manuscript presented in an intelligible fashion and written in standard English?

Reviewer #1: Yes

Reviewer #2: Yes

6. Review Comments to the Author

Reviewer #1: The authors have responded to all the reviewers' comments, and the new version is much clearer. The introduction of the experiment is much more complete, the calculation of the score better explained, and the analysis of the results more detailed. However, I have some questions about the new sections added with respect to the previous version: Scoring and Breakdown participant of individual participant involvement.

Scoring

1. The authors state that both accuracy and precision are real numbers, so that they can take negative values. I find this a bit counter-intuitive to understand. What would a negative accuracy or precision mean? What are their consequences on the outcome of the game?

2. The authors use values of 0.1 and 70 for the weights associated with accuracy and precision, respectively. Why do you use these particular two values? Is it an arbitrary decision? How would the game's outcome and even the conclusions change if you used another range of values for the analysis?

Breakdown of individual participant involvement

- The analysis presented in Table 5 that concludes there is no "learning" during the rounds is a bit sparse. These results show that the mean value may be comparable but don't explain the distribution of values, quartiles or outliers. I suggest performing the same analysis but presenting the results for each round in a boxplot so that the information presented is more visual and complete.

Other changes

- I would also suggest the authors to change the notation dx to d, since this variable represents the distance in the plane, and not only in the x axes.

- Please revise grammar typos in lines 163 and 164.

Reviewer #2: (No Response)

7. PLOS authors have the option to publish the peer review history of their article (what does this mean?). If published, this will include your full peer review and any attached files.

Reviewer #1: **Yes: **Diego Escribano

Reviewer #2: No

---

## [Editor Report · Acceptance letter]

27 Sep 2022

PONE-D-22-10361R1 

Person-to-person opinion dynamics: an empirical study using an online game 

Dear Dr. Adams:

I'm pleased to inform you that your manuscript has been deemed suitable for publication in PLOS ONE. Congratulations! Your manuscript is now with our production department. 

Kind regards, 

on behalf of

Dr. José Manuel Galán 

Academic Editor

PLOS ONE